# Causal role for the subthalamic nucleus in interrupting behavior

**Kathryn H Fife[1], Navarre A Gutierrez-Reed[2], Vivien Zell[3], Julie Bailly[3], Christina M Lewis[4], Adam R Aron[4], Thomas S Hnasko[3]***

[1]Neurosciences Graduate Program, University of California, San Diego, San Diego, United States; [2]Biomedical Sciences Graduate Program, University of California, San Diego, San Diego, United States; [3]Department of Neurosciences, University of California, San Diego, San Diego, United States; [4]Department of Psychology, University of California, San Diego, San Diego, United States

**Abstract** Stopping or pausing in response to threats, conflicting information, or surprise is fundamental to behavior. Evidence across species has shown that the subthalamic nucleus (STN) is activated by scenarios involving stopping or pausing, yet evidence that the STN causally implements stops or pauses is lacking. Here we used optogenetics to activate or inhibit mouse STN to test its putative causal role. We first demonstrated that optogenetic stimulation of the STN excited its major projection targets. Next we showed that brief activation of STN projection neurons was sufficient to interrupt or pause a self-initiated bout of licking. Finally, we developed an assay in which surprise was used to interrupt licking, and showed that STN inhibition reduced the disruptive effect of surprise. Thus STN activation interrupts behavior, and blocking the STN blunts the interruptive effect of surprise. These results provide strong evidence that the STN is both necessary and sufficient for such forms of behavioral response suppression.

*For correspondence: thnasko@ucsd.edu

**Competing interests:** The authors declare that no competing interests exist.

## Introduction

The subthalamic nucleus (STN) is a small structure with large functional significance for behavioral response control, decision-making, and clinical neuromodulation. The STN is composed principally of excitatory projection neurons (*Barroso-Chinea et al., 2007*; *Hammond et al., 1978*; *Smith and Parent, 1988*). It serves as a key node along the striatal indirect pathway, receiving inhibitory input via the external segment of the globus pallidus (GPe) (*Magill et al., 2004*; *Parent and Hazrati, 1995*). The STN projects broadly within the basal ganglia, sending dense projections to the internal globus pallidus (in rodents called entopeduncular nucleus, EP), substantia nigra pars reticulata (SNr), and reciprocal projections to GPe. In primates, damage to the STN is associated with uncontrolled voluntary movements (hemiballismus) (*Hamada and DeLong, 1992*). In rodents, genetic disruption of excitatory output from the STN via partial knockout of the vesicular glutamate transporter 2 (VGLUT2) induces hyperlocomotion (*Schweizer et al., 2014*), and lesions induce impulsive responding (*Baunez and Robbins, 1997*; *Eagle et al., 2008*; *Uslaner and Robinson, 2006*).

Many studies have shown that the STN is activated during specific tasks that require stopping or pausing behavioral output by suppressing pre-potent response tendencies. For example, the STN is activated by signals to stop an initiated response as shown by human fMRI (*Aron and Poldrack, 2006*), local field potential recording (*Ray et al., 2012*; *Wessel et al., 2016a*), and single-unit recordings in humans (*Bastin et al., 2014*; *Benis et al., 2016*), non-human primates (*Isoda and Hikosaka, 2008*), and rodents (*Schmidt et al., 2013*). The STN is also activated by the need to delay responding during conflict (reviewed by *Zavala et al., 2015*) or in response to surprising events (*Wessel et al., 2016b*). Anatomical and computational models propose that when stop signals,

conflict signals, or surprising events recruit the STN it activates basal ganglia output nuclei, transiently inhibiting thalamocortical drive (*Bogacz and Gurney, 2007*; *Mink, 1996*; *Wiecki and Frank, 2013*). This induces an outright stop, or a delay to buy time for more evidence to accumulate about the correct course of action, such as which response to perform in the case of conflict, or perhaps how to take evasive action after surprising or threatening stimuli (*Jahanshahi et al., 2015*; *Wessel and Aron, 2017*; *Zavala et al., 2015*). Yet there is a striking lack of evidence showing the STN has a causal role in such forms of response control. While STN lesions did affect behavior in stop signal tasks, the effects were modest and not specific to stopping (*Eagle et al., 2008*; *Obeso et al., 2014*). Therefore, we here used optogenetics to selectively stimulate excitatory STN projection neurons with fast temporal precision to test the effects on behavior.

## Results

### Optogenetic activation of STN excites output nuclei

Using *Slc17a6-Cre* (VGLUT2-Cre) knock-in mice and Adeno-associated virus (AAV) vectors we expressed Channelrhodopsin-2 fused with yellow fluorescent protein (ChR2:YFP) or YFP alone (control) in STN glutamate cells. Prior to behavioral experiments we verified the approach. First, we used histology to demonstrate effective targeting of STN. We observed dense YFP label in STN (*Figure 1A*) and its major terminal regions including the GPe (*Figure 1B*), EP (*Figure 1C*), and SNr (*Figure 1D*), with some spread to the parasubthalamic nucleus and the ventromedial thalamus. Second, acute slice electrophysiology experiments confirmed optogenetic control over these circuits. Cell-attached recordings showed that STN neurons expressing ChR2 increased firing in response to 10 ms flashes of blue light at 40 Hz (*Figure 1E*) or to a single 50 ms pulse (*Figure 1F*). In both experiments firing persisted for the duration of the stimulus, followed by apparent brief rebound inhibition. To test connectivity, we next recorded light-evoked responses from post-synaptic cells juxtaposed to fluorescent ChR2-expressing STN terminals in the GPe, EP, and SNr –revealing excitatory post-synaptic currents (EPSCs) in each of these targets (*Figure 1—figure supplement 1A–D*). EPSCs that were assessed pharmacologically showed DNQX sensitivity, confirming broad glutamate-mediated postsynaptic effects (*Figure 1—figure supplement 1A–C*). To test whether stimulation of STN inputs could increase firing in postsynaptic targets we applied 40 Hz stimulation for 5 s while recording action potentials from GPe, EP, and SNr, observing a mean 2- to 4-fold increase in firing in each region (*Figure 1—figure supplement 1E–G*). Finally, to ensure our targeting strategy was effective in activating STN neurons and their postsynaptic partners in vivo, photostimulus trains were applied 90 min prior to sacrifice using either 0.5 mW or 10 mW light. This led to robust Fos expression in STN (*Figure 1G–I*), and in STN target regions (*Figure 1—figure supplement 1H–M*). Together, these results demonstrate that we can effectively use ChR2 to broadly activate STN output.

### Activation of STN interrupts or pauses behavior

To test whether photostimulation of the STN is sufficient to interrupt a recently initiated motor sequence, we employed a licking task (*Figure 2A*). ChR2 was expressed unilaterally in VGLUT2-Cre STN neurons, optic fibers were implanted just dorsal to the injection site, and expression and placements were assessed posthoc (*Figure 2B* and *Figure 2—figure supplement 1A*). For the licking task, mice were provided limited daily access to sweetened strawberry milk in lickometer-equipped chambers. After habituation, a laser delivered brief STN photostimulation following the second lick on a subset of self-initiated bouts (*Figure 2A*). We defined a bout as two or more licks with ≤750 ms inter-lick intervals, which included >92% of all licks from spontaneously licking mice (*Figure 2—figure supplement 2*). Unilateral STN photostimulation (10 mW, 10 pulses, 40 Hz) in ChR2-expressing mice caused a significant increase in the number of bouts containing precisely two licks (*Figure 2C*) and a dramatic reduction in bout length compared to YFP controls (*Figure 2E,F*). Similar results were observed using 20-fold less power (0.5 mW) or with bilateral stimulation (*Figure 2—figure supplement 1B,C*). Strikingly, a single 50 ms pulse delivered subsequent to the second lick in a bout showed the same effect (*Figure 2D,G* and *Video 1*). Further, when assessing the subset of bouts where 50 ms stimulation was triggered by the second lick but did not provoke interruption, we found a rightward shift in the inter-lick interval (ILI) distribution between the second and third licks

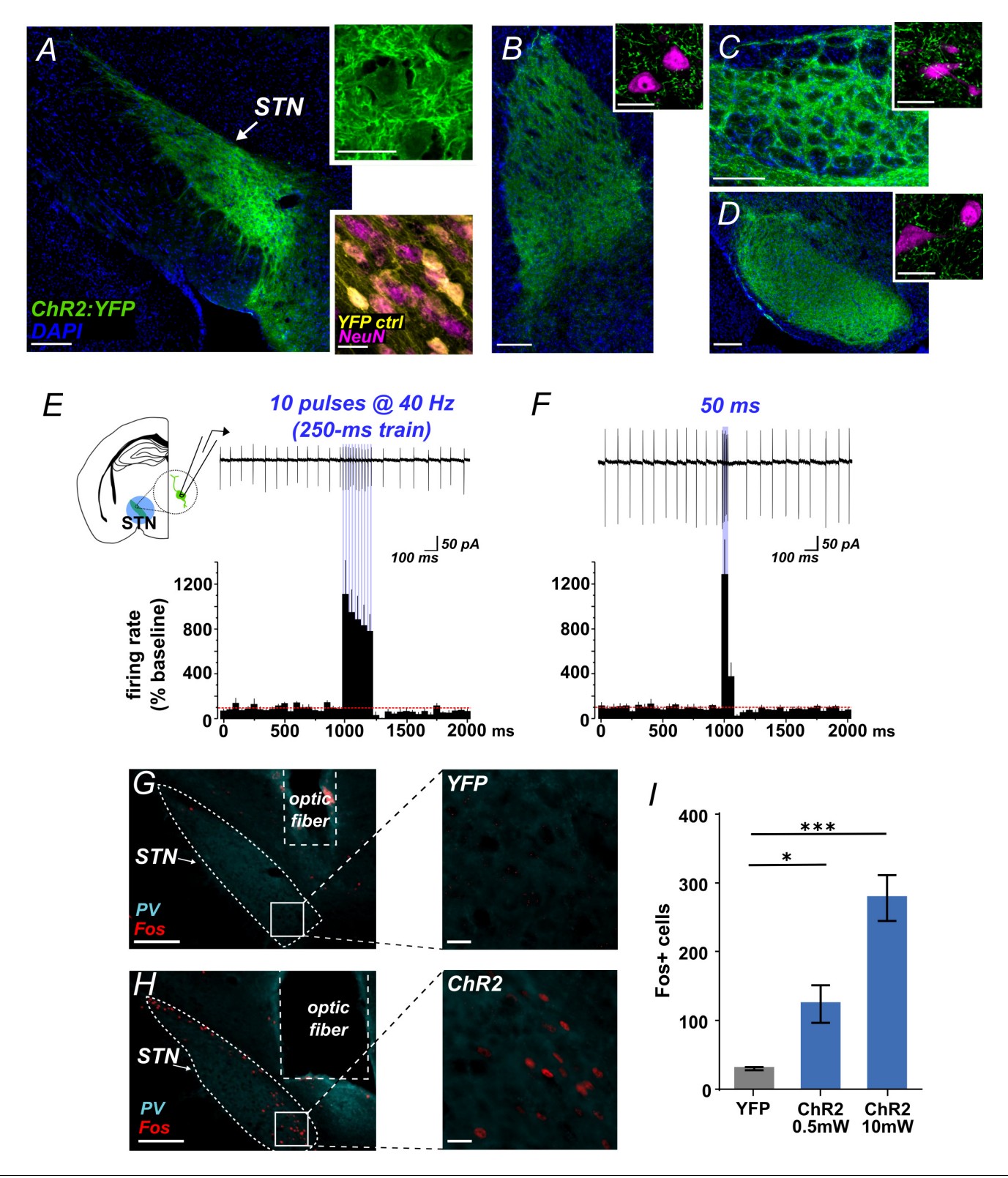

**Figure 1.** Functional photoactivation of STN projection neurons. (**A**) Image of coronal sections showing native ChR2:YFP fluorescence (green) in VGLUT2-Cre STN neurons; scale, 200 μ. High-magnification insets of ChR2:YFP (top) or YFP control (bottom) expression in STN cell bodies with co-labeling for nuclear marker (DAPI, blue or NeuN, pink); scale, 20 μ. Images through STN terminal fields in (**B**) GPe, (**C**) EP, and (**D**) SNr; scales as in A. (**E**) Example cell-attached recordings of action potentials from ChR2-expressing STN neuron in response to 10 ms blue light pulses at 40 Hz, or (**F**) a single

*Figure 1 continued on next page*

*Figure 1 continued*

50 ms blue light pulse. Histograms show % change in firing rate from a 1 s pre-stim baseline, n = 6 cells. (G) Example low- and high-magnification images show Fos immunolabeling in STN of YFP-control or (H) ChR2-expressing mice following in vivo photostimulation of STN (10 mW). Parvalbumin (PV) was used to delineate STN; scale 200 μ and 20 μ. (I) Fos-labeled cells are more abundant in ChR2:YFP-expressing STN compared to YFP control; n = 3-4 mice; unpaired t-test: t = 3.4, *p<0.05; t = 8.8, ***p<0.001.

The following figure supplement is available for figure 1:

**Figure supplement 1.** Functional photoactivation of STN targets.

(*Figure 2H*). In particular, STN activation reduces the occurrence of licks with ILIs between 100–200 ms (*Figure 2I*), resulting in a significant increase in the mean median ILI (*Figure 2J*), indicative of a behavioral delay, or pause.

We also examined the inter-bout interval (IBI), comparing the interval between bouts following laser trials versus non-laser trials. We found a significant shift toward longer IBI length following STN-induced behavioral interruption (*Figure 2—figure supplement 1D,E*). The reduced occurrence of short IBIs may indicate that STN activation induces a 'hard' interrupt, or a behavioral shift away from serial bouts of licking. Importantly, the ChR2 group showed no reduction but rather a (presumably compensatory) increase in the number of self-initiated bouts and licks (*Figure 2—figure supplement 1F,G*), suggesting that behavioral interruption is not explained by an aversive effect. Thus, STN stimulation rapidly and potently interrupts or pauses licking in a manner consistent with implementing a behavioral stop/pause.

## STN inhibition reduces the interruptive effect of surprise

While the ChR2 data clearly show that activation of STN disrupts a recently initiated motor sequence, the underlying mechanism is not clear; stimulation could have initiated an STN-mediated stop/pause (as per our hypothesis) or it could induce an alternate behavior or sensation that disrupts licking as a secondary consequence. We thus designed a behavioral assay that could more specifically point to a causal role for the STN in stopping/pausing. We hypothesized that because surprising events activate the STN (*Wessel et al., 2016b*) and interrupt licking (*O'Connor et al., 2015*), then inhibiting STN glutamate neurons using Halorhodopsin (Halo) (*Figure 3A,B* and *Figure 3—figure supplement 1A–E*) should attenuate the interruption induced by such events. To determine whether laser-mediated STN inhibition on its own induced any effects on licking, the second lick in a bout triggered a 1 s green laser pulse; we saw no change in lick pattern (*Figure 3C* and *Figure 3—figure supplement 1F,G*).

To determine the STN's role in surprise-induced behavioral inhibition, we developed an assay that used a combination of sound/light stimulus as a surprising event. A pilot experiment on untreated mice showed that, when triggered by the second lick in a bout, the surprising event potently interrupted behavior; moreover, mice habituated over the course of several sessions as would be expected by the waning of surprise (*Figure 3—figure supplement 2* and *Video 2*).

We next used this assay to test the effects of bilateral STN inhibition on surprise-induced interruption. Here, surprise was presented on 50% of self-initiated bouts, following the second lick and with a 100 ms delay. For half of these (25% of all trials), the laser was activated by the first lick in the bout without delay, thus preceding the sound/light stimulus by ~200 ms on average (*Figure 3D* and *Figure 2—figure supplement 2A*). As expected, surprise (without laser) interrupted licking, increasing the proportion of bouts with ≤3 licks for both YFP- and Halo-expressing groups (*Figure 3E*) and shifting the response distribution toward fewer licks per bout (*Figure 3F–G*) [NOTE: we assess ≤3 licks due to the 100 ms delay following the second lick]. Critically, however, on bouts where surprise was *preceded* by laser, we found a reduction in the number of short bouts in the Halo group compared to YFP control mice (*Figure 3E* and *Video 3*). This effect was observed repeatedly, with STN inhibition blunting the interruptive effects of surprise across several similar experiments and multiple cohorts (*Figure 3—figure supplement 1H*). These data show that STN inhibition reduced the disruptive effect of surprise on licking behavior, indicating that STN activity is necessary for normal surprise-induced interruption.

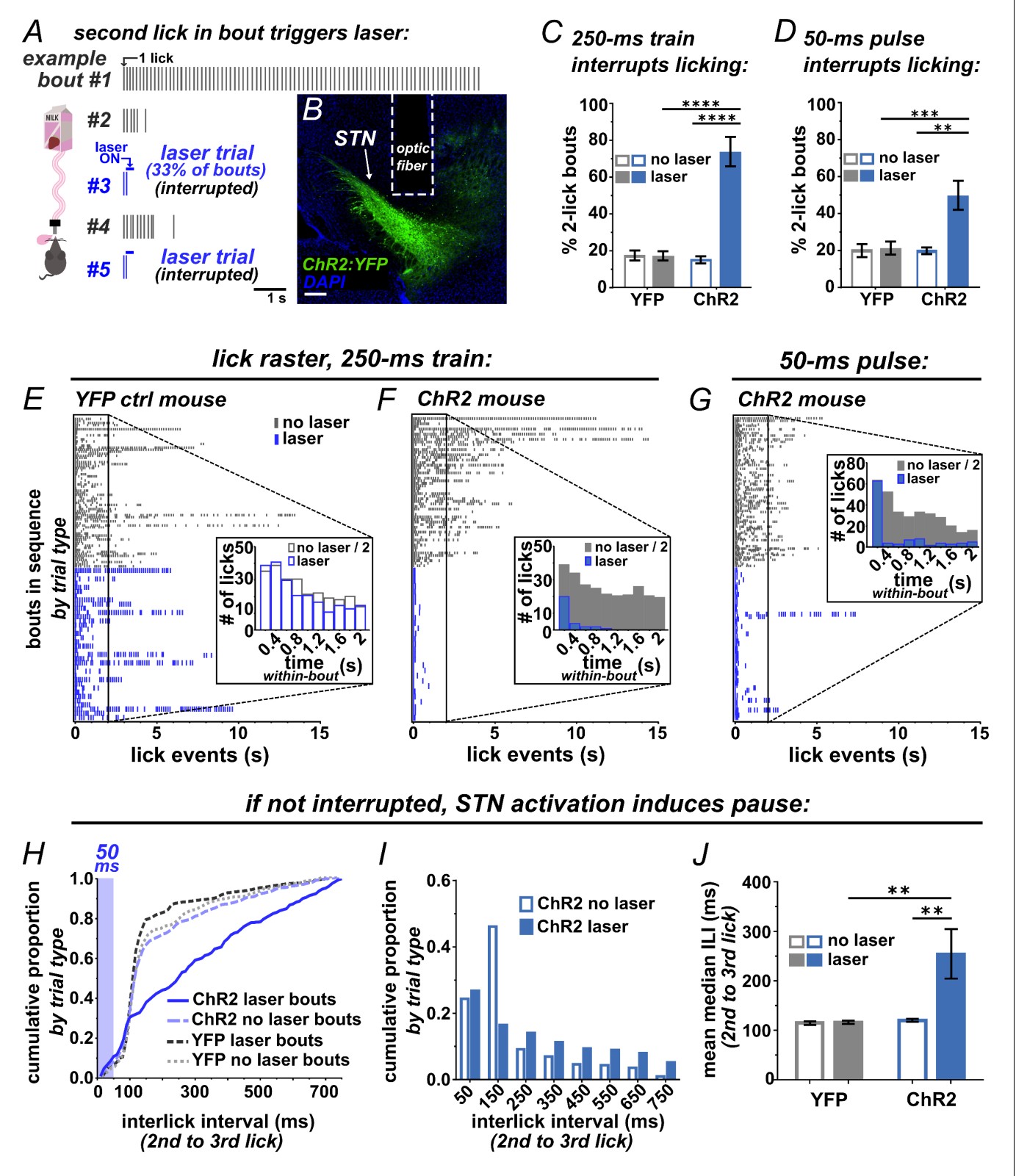

**Figure 2.** Brief optogenetic activation of STN rapidly interrupts licking. (**A**) Schematic of task, mice were provided 30 min daily access to palatable strawberry-milk and licks and bouts are recorded using contact lickometers. On one third of bouts within a given session, blue light photostimulation was delivered in response to the second lick of the bout. Vertical dashes represent licks within bouts of example animal. Horizontal dashes in laser trials represent timing of light delivery. (**B**) Coronal section showing unilateral ChR2:YFP expression in STN and optic fiber placement; scale 200 μ. (**C**) In

*Figure 2 continued*

ChR2:YFP mice, but not YFP controls, laser photostimulation (ten 10 ms 10 mW pulses at 40 Hz) increased the fraction of bouts that stop at precisely 2 licks (n = 8 YFP, n = 10 ChR2 mice; RM-ANOVA, treatment x stimulus interaction $F_{1,16}$ = 31.6, p<0.0001; Sidak posthoc ****p<0.0001. (D) Similar results were observed with animals subjected to a single 50 ms photostimulus (n = 8 YFP, n = 10 ChR2 mice, RM-ANOVA, treatment x stimulus interaction $F_{1,16}$ = 7.0, p<0.05; Sidak posthoc ***p<0.001, **p<0.01. (E) Example raster plots of licks within bout from YFP-control or (F, G) ChR2-expressing mice; insets show frequency distribution of licks in first 2 s of all bouts [Note that 'no-laser' values were divided by two to account for 2:1 ratio of trial type]. (H) In bouts that were not interrupted (>2 licks), 50 ms photostimulation led to a shift in the ILI distribution between the second and third licks in ChR2 mice; Kolmogorov-Smirnov (KS), p<0.0001. (I) This pause in licking was most apparent between 100–200 ms (100 ms bins) and (J) led to an increase in the mean median ILI between second and third licks; n = 8 YFP, n = 10 ChR2 mice; RM-ANOVA, treatment x stimulus interaction $F_{1,16}$ = 5.4, p<0.05; Sidak posthoc **p<0.01.

The following figure supplements are available for figure 2:

**Figure supplement 1.** Optic fiber placements and effect of STN photostimulation on licking.

**Figure supplement 2.** Patterns of self-initiated spontaneous licking in mice.

## Discussion

We showed that optogenetic stimulation of the STN, delivered after the second lick in a bout, rapidly interrupted or paused licking. The interruption occurred even when the optogenetic stimulation was brief (50 ms) or delivered at low power (0.5 mW), supporting the specificity of this manipulation and suggesting that behavior can be rapidly and potently interrupted by transient increases in STN output. While these observations are consistent with activating an STN-mediated stop/pause system, it is also possible that STN stimulation induced independent effects, such as inducing new fragments of behavior that competed with licking. Additionally, without in vivo recordings from STN neurons during behavior, we cannot be certain how optogenetic manipulation changed STN activity.

We therefore tested putative STN engagement in stopping/pausing in a different way. We took advantage of a recent finding that the STN is activated by surprising events in humans (*Wessel et al., 2016b*). We reasoned that if this is also the case in mice then optogenetic inhibition of the STN may reduce the interruptive effect of surprise on behavior. That is what we found. First, when the surprising event (sound and light) occurred 50 or 100 ms after the second lick in a bout, it had a clear interruptive effect on licking. Interestingly, this interruption was similar to that elicited by STN activation in the ChR2-expressing mice. Second, STN inhibition in the Halo-expressing mice strongly and reproducibly mitigated the interruptive effects of surprise on licking when compared to YFP-expressing control mice. Importantly, absent surprise, 1 s STN inhibition did not alter licking behavior, this despite previous studies linking STN lesions to hemiballismus and dyskinesia (*Hamada and DeLong, 1992*). These observations suggest that involuntary movements result from the sustained rather than acute loss of STN output, perhaps due to compensatory changes in basal ganglia circuitry.

In sum, our data demonstrate that acute STN activation interrupts behavior, and acutely blocking the STN diminishes the impact of surprise on behavior. Together, these data provide causal evidence of a role for the STN in stopping/pausing. The results validate a large empirical literature showing that STN activity is elicited by stop signals, switch signals, and decision conflict (*Wessel and Aron, 2017*; *Jahanshahi et al., 2015*; *Wiecki and Frank, 2013*; *Zavala et al., 2013*). Our data strongly suggest that increased STN activity in response to such signals observed in previous studies is not epiphenomenal, or correlated with some other variable such as arousal or effort, but instead reflects an implementation of stopping/pausing driven by STN recruitment.

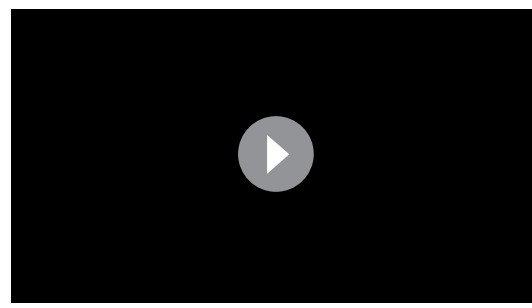

**Video 1.** Movie showing example of STN-activation interrupting licking. Related to *Figure 2*. Note that light-leakage was blocked during data acquisition, but left unblocked as a visual aid for this example movie.

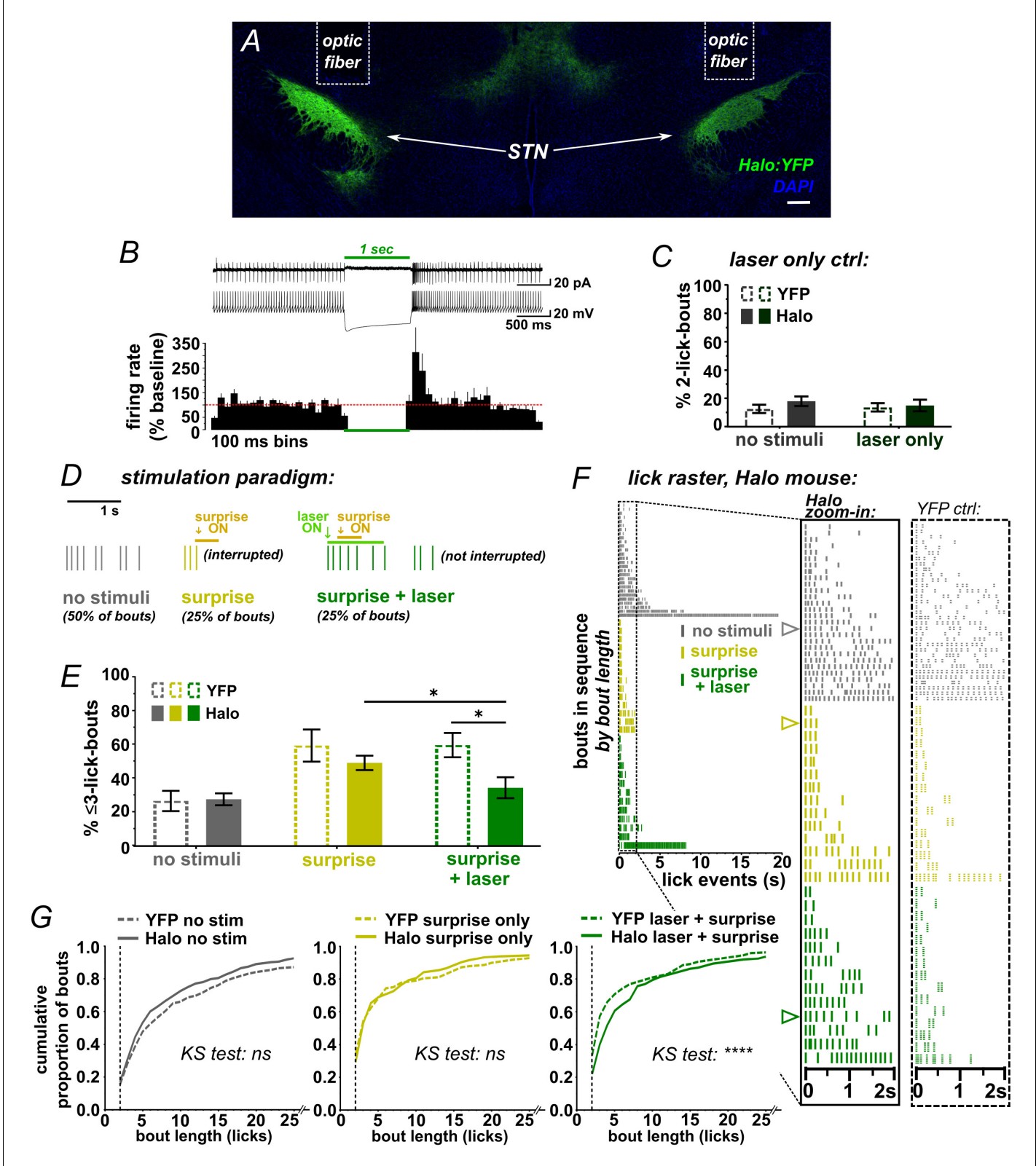

**Figure 3.** STN inhibition reduces the impact of surprise on interrupting licking. (**A**) Image of coronal section showing bilateral expression of eNPh3.0: EYFP (Halo) in STN. (**B**) Cell-attached (top trace) or whole-cell current-clamp (bottom trace) recordings from Halo-expressing STN neurons. Histogram shows % change in firing rate from a 2 s baseline; n = 7 cells. (**C**) Green laser photoinhibition delivered alone (following the second lick in a bout on 50% of bouts) did not affect licking in YFP-control or Halo-expressing mice; n = 8 YFP and 11 Halo mice. (**D**) Schematic of task. On 50% of bouts, the

*Figure 3 continued on next page*

*Figure 3 continued*

second lick triggers a 100 ms delay followed by visual and auditory 'surprise' stimuli to disrupt licking behavior. On 25% of bouts the surprise is preceded by green laser to photo-inhibit, with the laser delivered on the first lick in a bout and ending 950 ms after surprise onset. (E) Plot showing the number of bouts ending with three or fewer licks is increased by surprise, but laser inhibition reduces the interruptive effect of surprise on licking in the Halo-expressing mice compared to controls; n = 7 YFP and 11 Halo mice; RM-ANOVA, stimulus effect $F_{2,32}$ = 24, p<0.0001; treatment x stimulus interaction $F_{2,32}$ = 5.3, p=0.01; Sidak posthoc *p<0.05, surprise vs no stim p<0.0001 (YFP) and p<0.001 (Halo). (F) Example lick raster from a Halo-expressing mouse, insets include data from a YFP-control mouse for comparison. Arrowheads in raster denote the bouts illustrated in panel D. (G) Cumulative probability plots comparing Halo- vs YFP-expressing mice bout length distributions without stimulus, with surprise-induced interruption, and with surprise plus laser. When compared to YFP controls, STN inhibition reduced the interruptive effects of surprise; n = 7 YFP, n = 11 Halo; KS = Kolmogorov-Smirnov, ****p<0.0001, ns = not significant.

The following figure supplements are available for figure 3:

**Figure supplement 1.** Optic fiber placements and effect of STN inhibition on licking.

**Figure supplement 2.** Surprise-induced interruption of licking.

Note that much previous work on the role of the STN in response inhibition tested its role on pre-potent actions, prior to the initiation of action, for example using Go/NoGo or stop-signal tasks. Here, however, we assessed the effect of the STN on a recently initiated action sequence, ~100–300 ms *after* the measurement of an action (lick bout), instead of *before* it. In other words, we found that STN activity can inhibit an ongoing action, but we did not directly test its role in inhibiting action initiation. Though this is an essential distinction from an experimental point of view, these concepts are conceptually overlapping, and we suppose rely on similar circuit mechanisms. We suppose that an STN-driven system for Stopping or Pausing in response to surprise or conflict signals, to interrupt or delay an action plan in order to consider new evidence, has ethological utility whether a new action sequence is being initiated, or is already underway.

These results also have wider implications. Recent studies have linked STN-mediated stopping to working memory decrements (*Wessel et al., 2016b*), suggesting neural circuit links between stopping behavior and cognition. Further, excess and aberrant activity in the STN may induce a pathological state of stopping and interruption, and explain why STN lesions or deep brain stimulation dramatically improve Parkinsonian motor deficits (*Brittain et al., 2014*; *Wichmann et al., 2011*). By providing causal evidence that brief activation of the STN is sufficient to interrupt or pause behavior, and that STN activity contributes to surprise-induced behavioral interruption, the current results put the functional role of the STN on a much firmer footing.

## Materials and methods

### Animals

Homozygous *Slc17a6-IRESCre/IRESCre* (VGLUT2-Cre) mice (*Vong et al., 2011*) obtained (The Jackson Laboratory, stock #016963), maintained in-house on a C57Bl/6 background, and used in accordance with guidelines established by the Institutional Animal Care and Use Committee at the University of California, San Diego. Mice were maintained on a 12:12 hr light-dark cycle in a temperature- and humidity-controlled environment, group-housed by sex in plastic cages (maximum five mice/cage) with lofts and cotton nestlets for enrichment, and food and water were available *ad libitum* unless specified. Both male and female mice (>6 wks) were used.

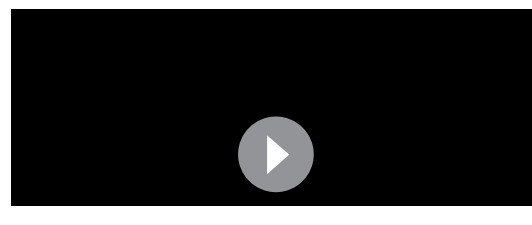

**Video 2.** Movie showing example of surprise-induced interruption of licking. Related to *Figure 3*.

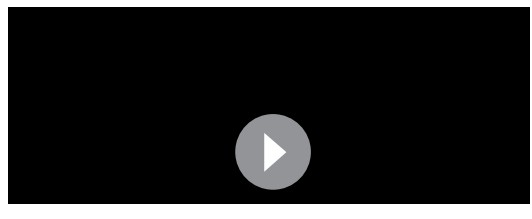

**Video 3.** Movie showing example of STN-inhibition blocking surprise-induced interruption of licking. Related to *Figure 3*. Note that light-leakage was blocked during data acquisition, but left unblocked as a visual aid for this example movie.

## Stereotactic surgery

Anesthetized VGLUT2-Cre mice (isofluorane, 2%) were placed in a stereotactic apparatus (David Kopf Instruments), a small incision was made on the scalp, the skull was leveled, a hole drilled above the STN. 400 nl of viral vector was injected at a rate of 50 nl/min using a custom-made stainless-steel 30 μm cannula (Plastics One) beveled at a 30° angle and connected to a micropump (World Precision Instruments) via back-filled polyethelene tubing (Becton Dickinson and Company). Cre-dependent expression of YFP-tagged Channelrhodopsin-2 (ChR2; H134R), YFP-tagged halorhodopsin (eNpHR 3.0), or YFP (controls) was achieved with rAAV1-EF1α-DIO-ChR2:YFP ($4 \times 10^{12}$ genomes/ml), rAAV5-EF1α-DIO-eNpHR3.0:YFP ($3 \times 10^{12}$), or AAV5-EF1α-DIO-EYFP ($3 \times 10^{12}$) for behavioral experiments. For a subset of the electrophysiological experiments we used rAAV1-EF1α-DIO-ChR2:mCherry ($2 \times 10^{12}$) and this is noted in the legends. All vectors were obtained from the University of North Carolina viral vector core. Injections were made bilaterally or, where noted, unilaterally into the left hemisphere to target the STN at the following coordinates (in mm relative to Bregma):±1.6 ML, −2.0 AP, −4.5 DV. After injection we waited 10 min before removing the injector to minimize backflow. For behavioral experiments, a custom-made 220 μm core optic fiber (Thorlabs, Inc., BFL37-200) connected to a ceramic ferrule (Precision Fiber Products, Inc., MM-FER2007C-2300) (*Yoo et al., 2016*; *Sparta et al., 2011*) was implanted dorsal to the injection site at: ± 1.6 ML, −2.0 AP, −4.35 DV. Optic fibers were stabilized in place using three skull screws (Plastics One, Inc., 00–96 × 1/16) and dental cement (Lang dental). Mice were given post-operative analgesic (Carprofen, Pfizer, 5 mg/kg s.c.), ophthalmic ointment was used to protect eyes during surgery, and betadine was used at incision site. Mice were allowed to recover for ≥3 weeks before subsequent assay.

## Histology and Fos

Following behavioral experiments, subsets of mice were stimulated in an open field for 30 min at 40 Hz for 5 s on, 10 s off (10 ms pulse width, 10 mW), and perfused 90 min thereafter. Mice were injected with a lethal dose of pentobarbital (Euthasol, Virbac 200 mg/kg i.p.) and an intra-cardiac perfusion was performed with ~10 ml ice-cold phosphate buffered saline (PBS) followed by ~50 ml freshly made ice-cold 4% paraformaldehyde (PFA, Electron Microscopy Sciences) using a peristaltic pump at ~6 mL/min. Brains were removed, post-fixed overnight at 4°C in PFA, transferred to 30% sucrose in PBS for 48 hr, flash-frozen in −30°C isopentane, and stored at −80°C. Brains were processed into 30 μm coronal sections using a cryostat (CM3050S, Leica) and sections were stored at 4°C in PBS containing 0.01% sodium azide. Sections from each animal were first examined for native fluorescence and implant site; two mice were excluded from the analysis due to optic fiber misplacement (*Figure 2—figure supplement 1A*). For immunohistochemistry sections underwent 3 × 5 min washes in PBS, then washed 3 × 5 min with PBS-Tx (0.02% Triton X-100), followed by 1 hr in 4% normal donkey serum (NDS) in PBS-Tx (block), each at room temperature with gentle agitation. Sections were then incubated in one or more of the primary antibodies (1:200 guinea pig anti-NeuN, Millipore, ABN90, RRID: AB_11205592;1:2500 rabbit anti-Fos, Calbiochem PC38, RRID: AB_2106755; 1:2000 mouse anti-parvalbumin, Millipore, MAB1572, RRID: AB_2174013); 1:2000 rabbit anti-GFP, Invitrogen, A11122, RRID: AB_221569) overnight at 4°C with gentle agitation. Note: anti-GFP antibody was only used for YFP control sections (*Figure 1A*, lower right); all ChR2 and Halo images show native fluorescence. Sections were then washed 3 × 5 min with PBS-Tx, incubated for >2 hr at RT in species appropriate cross-absorbed donkey antibodies conjugated to Alexa594 or Alexa647 (1.5 μg/mL, Jackson ImmunoResearch), washed 3 × 5 min in PBS, and mounted on to glass slides with Fluoromount-G mounting medium (Southern Biotech) ± DAPI (Roche, 0.5 μg/mL).

For Fos quantification, images were acquired using an inverted epifluorescence microscope (Zeiss AxioObserver) with a motorized stage and Zen (Zeiss) software through a 20X (0.8 NA) objective with identical acquisition settings across samples. Cell counting was performed in NIH-image (Fiji) using identical gain and offset settings between samples. For STN cell counts PV label was used to define STN boundaries, tiled images were made to encompass the entire region, and three sections through the STN (between Bregma −1.70 to −2.18) were counted per animal. High-magnification images were acquired using identical acquisition settings across samples with an Olympus confocal microscope (Fluoview FV1200) using a 60X (1.35 NA) objective or a Zeiss AxioObserver with Apo-Tome using a 63X objective (1.4 NA).

## ChR2-evoked lick interrupt

Mice were habituated by handling daily for 5 days prior to testing, and tethered to a patch cable on the last 2 days of habituation. Sweetened strawberry-flavored milk (9 g dry milk and 6.25 g straw-berry Nesquik per 100 ml drinking water) was used to encourage vigorous drinking in *ad libitum* fed mice, and was provided by hand during the habituation period. Operant chambers (Med Associates, ENV-352–2W) in sound-isolated boxes illuminated by 2 LED light strips and equipped with a retract-able lickometer sipper were used for testing. Licks were detected and recorded by a PC running Med-PC IV (Med Associates), with code written in MEDState notation. Mice were tethered to the laser via their implanted fiber/ferrule via patch cables coupled to an optical commutator (Doric Lenses, Canada) and placed into the chamber. Test sessions were 30 min and conducted during the light cycle. Following the second lick in a self-initiated bout, and with 33.3% probability, the computer triggered a DPSS laser (473 nm, Shanghai or OEM Laser) to deliver photostimulation. A lick bout was defined as two or more consecutive licks with an inter-lick interval of ≤750 ms. Photostimu-lation was driven by custom-controlled Arduino stimulus generators and consisted of 10 mW (~80 mw/mm$^2$ at implanted fiber tip, calibrated per animal) continuous stimulation for 50 ms, or ten 10 ms pulses at 40 Hz.

## Surprise-evoked lick interrupt

The same operant chambers and software were used as in the ChR2-evoked lick-stop task (see above). Mice were handled and exposed to strawberry milk for 5 days. Mice were then habituated to the operant chambers for 15 days; during habituation, mice had free access to strawberry milk; beginning on the third day mice were tethered to the laser. Testing sessions were 60 min and con-ducted during the dark cycle. A criterion of ≥6 bouts for each trial type was pre-determined before testing; any mice that did not reach this criterion were excluded from analyses. Surprise stimuli (0.5 s) included activation of an overhead houselight and speaker delivering white noise (10 dB) on 25% of bouts, triggered by the second lick in a bout with a 100 ms delay. On another 25% of trials, con-tinuous 10 mW green light (532 nm DPSS, Shanghai Laser) was triggered immediately following the 1$^{st}$ lick, which then persisted until 1050 ms after the second lick. Following the second lick, a 100 ms delay was triggered, after which the same surprise stimuli were delivered. On trials where the mouse did not reach a second lick (as defined by our bout criteria) the laser was shut off after 750 ms and these trials were not considered as bouts or included in analyses. Modeling clay was applied at the junction of the patch cable to block all light leakage. Sessions were repeated up to four times to reach criterion and pooled. During the laser-only test (*Figure 3C*) 1 s of continuous 10 mW light was delivered immediately following the second lick on 50% of bouts. Iterations of the surprise+laser main effect (*Figure 3—figure supplement 1H*) used varied conditions, including the timing of laser onset, unilateral versus bilateral inhibition, and whether modeling clay was applied at the junction of the patch cable to block all light leakage. In these experiments a 1 s laser pulse was triggered by the second lick in the bout (rather than the first) and surprise was delayed 50 ms (rather than 100 ms) after the second lick.

## Ex vivo electrophysiological recordings

Mice (7–11 weeks) were deeply anesthetized with pentobarbital (200 mg/kg i.p.; Virbac) and per-fused intracardially with 10 ml ice-cold sucrose-based artificial cerebrospinal fluid (ACSF) containing (in mM): 75 sucrose; 87 NaCl, 2.5 KCl, 7 MgCl$_2$, 0.5 CaCl$_2$, 1.25 NaH$_2$PO$_4$, 25 NaHCO$_3$ and continu-ously bubbled with carbogen (95% O$_2$, 5% CO$_2$). Brains were extracted and 200 μm coronal slices

were cut in sucrose-ACSF using a Leica Vibratome (vt1200). Slices were transferred to a perfusion chamber containing ACSF at 31°C (in mM): 126 NaCl, 2.5 KCl, 1.2 $MgCl_2$, 2.4 $CaCl_2$, 1.4 $NaH_2PO_4$, 25 $NaHCO_3$, 11 glucose, continuously bubbled with carbogen. After at least 45 min recovery, slices were transferred to a recording chamber continuously perfused with ACSF (2–3 ml/min). Patch pipettes (3.5–5.5 MΩ) were pulled from borosilicate glass (King Precision Glass) and for voltage-clamp recordings filled with internal recording solution containing (in mM): 120 $CsCH_3SO_3$, 20 HEPES, 0.4 EGTA, 2.8 NaCl, 5 TEA, 2.5 Mg-ATP, 0.25 Na-GTP, at pH 7.25 and 285 ± 5 mOsm. For cell-attached and current-clamp recordings of ChR2- and eNpHR3.0-expressing STN neurons, a potassium-based recording solution was used (in mM): 123 $KCH_3SO_3$, 10 HEPES, 0.2 EGTA, 8 NaCl, 2.5 Mg-ATP, 0.25 Na-GTP, at pH 7.25 and 280 ± 5 mOsm.

Fluorescent STN neurons and terminals were visualized by epifluorescence and visually-guided patch recordings were made using infrared-differential interference contrast (IR-DIC) illumination (Axiocam MRm, Examiner.A1, Zeiss). ChR2 was activated by flashing blue light through the light path of the microscope using a light-emitting diode (LED460, Prizmatix) under computer control. Excitatory postsynaptic currents (EPSCs) were recorded in whole-cell voltage clamp or action potentials were recorded in cell-attached mode (Multiclamp 700B amplifier, Molecular Devices), filtered at 2 KHz, digitized at 10 KHz (Axon Digidata 1550, Molecular Devices) and collected on-line using pClamp 10 software (Molecular Devices). Series resistance and capacitance were electronically compensated prior to whole-cell recordings. Estimated liquid-junction potential was 12 mV and left uncorrected. Series resistance was monitored during recordings and cells that showed >25% change during recordings were considered unstable and discarded from analyses. To assess the effects of ChR2 activation and Halo inhibition in the STN we used cell-attached or current-clamp and assessed responses to a single 50 ms pulse, or 10 10 ms pulses (40 Hz) of blue light (ChR2), or 1 s pulse of green light (Halo), delivered every 55 s and 3 responses were averaged. For post-synaptic firing rates, cell-attached recordings were averaged over the 5 s before, 5 s during photostimulation (40 Hz, 200 pulses, 5 ms pulse width) and 5 s after; 3 responses were averaged per neuron. Average effect is also shown as timeplot histograms where each bar (200 ms bin) is relative to the baseline (baseline has been calculated as the average of all the pre-photostimulation bins for each neuron separately). To assess EPSCs, neurons were held in voltage-clamp at −60 mV, a single pulse (5 ms) photostimulus was applied every 60 s and 10 photo-evoked EPSCs were averaged per neuron per condition. DMSO stock solution of DNQX (Sigma) was diluted 1000-fold in ACSF and bath applied at 10 μM. Current sizes were calculated by using peak amplitude from baseline.

## Statistics

Data values are presented as means ± SEM unless noted and subjected to unpaired or paired t-test or repeated-measures ANOVA followed by Sidak post hoc analysis where appropriate (Prism). Frequency distributions were compared using the Kolmogorov-Smirnov test (Prism). Statistical significance was set at $p < 0.05$.

## Acknowledgements

This work was supported by an NSF GRFP (KF), the NIDA-INSERM fellowship program (VZ), and NIH grants NS087496 & DA036612 (TSH). We thank Alex Johnson and Elizabeth Souter for assistance breeding mice, Byungkook Lim for access to microscope, and Doug Nitz, Tina Gremel, Jon Heston, and Nick Hollon for comments on the manuscript.

## Additional information

### Funding

| Funder | Grant reference number | Author |
| --- | --- | --- |
| National Science Foundation | GRFP | Kathryn H Fife |
| National Institute on Drug Abuse | NIDA-INSERM Postdoctoral Fellowship | Vivien Zell |
| National Institutes of Health | R21NS087496 | Thomas S Hnasko |

| National Institutes of Health | R01DA036612 | Thomas S Hnasko |

The funders had no role in study design, data collection and interpretation, or the decision to submit the work for publication.

### Author contributions

KHF, Conceptualization, Methodology, Software, Validation, Formal analysis, Investigation, Data curation, Writing—original draft, Writing—review and editing, Visualization, Project administration, Funding acquisition; NAG-R, Conceptualization, Methodology, Software, Formal analysis, Writing—review and editing, Investigation, Visualization; VZ, Formal analysis, Investigation, Data curation, Writing—review and editing, Visualization,Funding acquisition; JB, Validation, Investigation; CML, Investigation; ARA, Conceptualization, Writing—original draft, Writing—review and editing, Project administration; TSH, Conceptualization, Methodology, Resources, Writing—original draft, Writing—review and editing, Visualization, Supervision, Project administration, Funding acquisition

### Author ORCIDs

Thomas S Hnasko, http://orcid.org/0000-0001-6176-8513

### Ethics

Animal experimentation: All procedures in this study were performed in accordance with guidelines established by the Institutional Animal Care and Use Committee (IACUC) at the University of California, San Diego. All protocols were pre-approved by IACUC before conducting experiments (Protocol #: S12080). All surgeries were performed under isofluorane anesthesia, mice were monitored daily for a minimum of 5 subsequent days following surgery, and every effort was made to minimize suffering.

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
