## [Decision Letter]

[Editors’ note: a previous version of this study was rejected after peer review, but the authors submitted for reconsideration. The first decision letter after peer review is shown below.]

Thank you for submitting your work entitled "Causal role for the subthalamic nucleus in interrupting behavior" for consideration by *eLife*. Your article has been reviewed by three peer reviewers, and the evaluation has been overseen by a Reviewing Editor and a Senior Editor. The reviewers have opted to remain anonymous.

Our decision has been reached after consultation between the reviewers. Based on these discussions and the individual reviews below, we regret to inform you that your work will not be considered further for publication in *eLife*.

Summary:

The authors have tested the idea that subthalamic nucleus (STN) plays an important role in interrupting or pausing on-going behavior by a surprising stimulus or threat. The authors first developed a task using self-initiated licking behavior. The authors first show that optogenetic activation of STN neurons was sufficient to pause a bout of licking. Furthermore, optogenetic inhibition of STN neurons reduced the disruptive effect of a salient light/sound stimulus on licking behavior, effectively lengthening lick bouts.

The reviewers thought that the task is a naturalistic behavior and simple yet elegant. Although the role of STN in interrupting behaviors is not very novel, it is important to test this idea experimentally. Overall, the reviewers thought that this study is important and potentially warrant publication in *eLife*.

However, the reviewers raised a number of concerns. In particular, it is important to quantify how halorhodopsin-mediated inhibition affected the spiking of STN neurons. Since this experiment will likely take >2 months, we reject this manuscript at least in its current form. However, if the authors can address the following essential points, we are happy to reconsider a new submission of this work.

Essential points:

1) Although the authors confirmed the effect of optogenetic activations via channelrhodopsin-2 (ChR2), they do not show any demonstration that halorhodopsin inhibited the activity of neurons in the subthalamic nucleus (STN). This is an important point to demonstrate in order to interpret the data. Were STN neurons purely inhibited? What about rebound excitations? Please address this issue by in vivo recording.

2) In Figure 1, the authors show the effect ChR2 stimulation in the STN. Spiking of the STN neurons seems essentially continuous during the stimulation train. Does this indicate that synchronous activation of STN neurons caused persistent reverberatory activity? Please show the data after the last pulse as well so that we can see the extent at which the effect of stimulation lasts after the termination of the stimulation.

3) Figure 1. If the idea is that STN can rapidly arrest behavior, why show target firing rate averaged over a 5s stimulation period? Presumably any relevant effect on target firing must be occurring within less than ~50ms or so – please show the time courses of target firing shortly after light onset instead.

4) In general, it would be important to show less processed behavioral data, which may give a clearer view of what is going on. For instance, in Figure 2, the authors show an idealized schematic of a regular lick bout, and then information on lick bout length. Please show actual lick bout timing. Examples of real bouts may be helpful, and also simply rasters and histograms of lick density over time (as if licks were spikes). Also, are the bouts very distinct from each other or is the mouse essentially licking all the time with occasional >750ms gaps that define bout boundaries?

5) Figure 3. For the surprise experiment, why use such a long laser stimulation pulse (1s) given that any relevant neural effects must be much faster? Is this related to the definition of a "bout" as involving <750ms inter-lick-intervals? In addition, it seems that it took a long time for STN neurons to be inhibited by the halorhodopsin-based inhibition, compared with the excitation by ChR2-based stimulation. This also points to the importance of characterizing the effect of halorhodopsin-based inhibition on STN neurons (point #1).

6) Figure 2—figure supplement 1. Why would green light alone (YFP group) tend to interrupt behavior when such an effect was not seen for blue light alone (YFP group) in the previous experiment?

7) Halo and YFP animals showed different lengths of lick bouts in the baseline (no surprise, no laser condition, although the difference was not significant) (Figure 3). Although surprise resulted in similar lick bout lengths in Halo and YFP mice (Figure 3), the difference in the baseline might be problematic. Combining these results, the main result in Figure 3 could be explained by a mixture of laser itself, individual biases, and the effect that the authors are looking for (the role of STN). The authors must discuss this.

*Reviewer #1:*

The authors examined behavioral effects of stimulating or blocking the subthalamic nucleus (STN) activity by applying optogenetics to mice. They found (1) STN activation interrupted or paused a self-initiated licking, and (2) STN silencing reduced the disruptive effect of surprise.

1) The authors only examined the effect of STN optogenetic activation by in vitro recording and c-fos immunohistochemistry. Results of in vivo electrophysiological recordings, how STN neurons and their targets are activated by light through the fiber optics placed above the STN, are necessary.

2) Moreover, they did not show any in vivo and in vitro electrophysiological results during STN silencing by Halorhodopsin. These data are indispensable.

3) How effective is the activation or inhibition of STN neurons? Silencing of the STN induces motor abnormal behaviors such as hemiballism. Did animals show abnormal behaviors, such as hemiballism or rotational behaviors during strong silencing of the STN?

4) Why did authors use different vectors between behavioral experiments and in vitro electrophysiological experiments? They should use the same vectors and examine the effectiveness by electrophysiological methods.

*Reviewer #2:*

To study the function of STN, the authors used a self-initiated bout of licking as an ongoing behavior that may be modulated by STN. This is a great choice because the behavior is natural and repeatable without any learning. The role of STN was examined by local activation and inactivation of STN neurons using optogenetic stimulation. Both of the data are critical for the conclusion that "STN is both necessary and sufficient for such forms of behavioral response suppression." The effects of STN activation are clear and convincing, but I have a question about the effects of STN inactivation, as shown below.

There is no clear evidence that photostimulation of STN in Halo-expressing mice inactivated STN neurons or their target neurons (GPe/EP/SNr), unlike the data shown for ChR2-expressing mice shown in Figure 1. This may be a bit tricky because STN neurons must be spontaneously active to see any effect on the target neurons. But if a prolonged stimulation (e.g., 1 s) is used (as in the behavioral experiment shown in Figure 3), the firing rates of the target neurons should decrease. Such data are important to proceed to the behavioral experiment.

I have two specific questions.

First, how quickly can STN neurons suppress ongoing behavior? This is important to adapt to the rapidly changing environment. To address this question, I have been checking the data in Figure 2. I assume that the blue window indicates the stimulation period. The effect of ChR2-laser diverged from the control around 100 ms after the onset of the stimulation. This looks explicit, but I am not convinced. My understanding is that the data lines are shown relative to the total number of licks within a bout of licks. Since the number of licks was smaller in ChR2-laser trials, the data lines are not presented based on the actual frequency of licks. I would simply show the cumulative number of clicks. Then, the blue line (ChR2-laser) would be lower, and the differentiation latency may be shorter than 100 ms. Another reason for asking this question is that the effects of the photostimulation on the STN-target neurons in GPe, EP, and SNr are fairly quick (Figure 1) (although I cannot see the actual latencies). If these target neurons respond, say, in 2-3 ms, I expect that the behavior would be suppressed much earlier than 100 ms.

Second, I have some questions about the data showing the effect of the inactivation of STN neurons (Figure 3). According to my understanding, the photostimulation started simultaneously with the 2nd lick, and then the surprising event started after 50 ms. Is this because the authors had tried several versions and found that this temporal order was most effective? Apparently, it took a long time for STN neurons to be inhibited by this Halo-based stimulation, compared with the excitation by ChR2-based stimulation. Relevant to this question: What was the latency of the behavioral suppression in response to the surprising event?

Other specific questions and comments:

Subsection “Optogenetic activation of STN excites output nuclei”, last paragraph: How did you define 'postsynaptic cells'?

In Figure 1, please indicate the time and EPSC amplitude for the example data. What was the latency of EPSC in response to the stimulation?

In Figure 1—figure supplement 1, does 'ChR2 excluded' mean that the data obtained with these stimulation sites were excluded? I presume that the stimulation effect was absent or weaker than the others. Such data may be important to support the conclusion: the stimulation affected STN, not other areas.

Data in Figure 1—figure supplement 2. Was the stimulation intensity 0.5mW or 10mW? There are some c-Fos labeled cells outside the presumed target areas. For example, I wonder if labeled cells outside SNr (K) are located in VTA.

Please indicate how many animals were used for each experiment.

Figure 2 indicate that the total number of bouts increased in ChR2-laser condition. Does this mean that the total number of licks increased? Any interpretation?

Figure 2—figure supplement 1 shows that the bilateral stimulation was less effective. Any reason?

Figure 3—figure supplement 1 suggests the non-selective effect of photostimulation which seems to act as another surprising event. Is it difficult to block the light from the head cap?

Figure 3—figure supplement 2 indicates that the behavioral suppression became weaker as the surprising event was repeated. I wonder if this is caused by the decrease in the sensitivity of STN neurons to the surprising event.

Was the experiment shown in Figure 3 started after the habituation shown in Figure 3—figure supplement 2? If so, why?

"Importantly, in the absence of the sound/light event we found that STN inhibition did not alter licking behavior compared to the YFP controls."

This is important, but I cannot find data.

While reading the manuscript, I had a difficulty in finding which video I should check.

*Reviewer #3:*

In this brief report Fife et al. present optogenetic results supporting the idea that the STN is involved in interrupting ongoing behavior; activation of STN tends to interrupt bouts of licking, while suppression of STN tends to prevent interruption of licking by surprising cues. Though limited in scope these results based on manipulations are a useful complement to the extensive literature on STN & stopping based on correlations. But it would be good to show less processed behavioral data, which may give a clearer view of what is going on.

1) (Figure 1) Spiking of the STN neurons seems essentially continuous during the stimulation train. This seems a bit strange – is synchronous stimulation of STN neurons causing persistent reverberatory activity? Please show us when it stops after the last pulse. In any case "spikes/stimulus" seems like an inappropriate measure, since it's not clear which spikes are being evoked by which stimulus.

2) (Figure 1). If the idea is that STN can rapidly arrest behavior, why show target firing rate averaged over a 5s stimulation period? Presumably any relevant effect on target firing must be occurring within less than ~50ms or so – please show the time courses of target firing shortly after light onset instead.

3) (More on Figure 1). Figure 1: probably unnecessary these days. Figure 1: not clear what is being shown – is this supposed to be ChR2 expression in STN cell bodies and in fibers (only) within STN targets? Probably better just to use Figure 1—figure supplement 1 as a main figure instead. Figure 1: scale bar scales are not shown or given in caption.

4) (Figure 2) We are shown an idealized schematic of a regular lick bout, and then information on lick bout length, but please show actual lick bout timing. Examples of real bouts may be helpful, and also simply rasters and histograms of lick density over time (as if licks were spikes). Also, are the bouts very distinct from each other or is the mouse essentially licking all the time with occasional >750ms gaps that define bout boundaries?

5) (Figure 2) If STN arrests licking, but mice adjust by increasing the number of bouts, how long does the arrest last?

6) (Figure 2) How fast is the closed-loop control? I.e. what is the time from 2nd lick onset to laser pulse onset?

7) (Figure 3) For the surprise experiment, why use such a long laser stimulation pulse (1s) given that any relevant neural effects must be much faster? Is this related to the definition of a "bout" as involving <750ms inter-lick-intervals?

8) (Figure 3) Schematic – it's not very clear what is what.

9) (Figure 2—figure supplement 1) Why would green light alone (YFP group) tend to interrupt behavior when such an effect was not seen for blue light alone (YFP group) in the previous experiment?

10) Discussion is a bit cursory. For example it would have been helpful to discuss the fact that the results here are based on interrupting an already-started action (if this is a fair description) compared to canonical results based on preventing action initiation at all.

[Editors’ note: what now follows is the decision letter after the authors submitted for further consideration.]

Thank you for resubmitting your work entitled "Causal role for the subthalamic nucleus in interrupting behavior" for further consideration at *eLife*. Your article has been favorably evaluated by Timothy Behrens (Senior Editor) and three reviewers, one of whom, Naoshige Uchida, is a member of our Board of Reviewing Editors.

The manuscript has been improved but there are some remaining issues that need to be addressed before acceptance, as outlined below:

Essential points:

1) Figure 1: It's not entirely clear what point is being made with the zoomed-in insets, and what the nuclear marker is in the YFPctrl inset.

2) Figure 1—figure supplement 1 not referenced in text?

3) Figure 2. The illustration of blue light coming out of fibers to interrupt licking is pretty, but better would be an unequivocal mark of exactly the period of blue light illumination.

4) Figure 3. The explanation for using "% 3-lick-bouts", rather than 2-lick-bouts as examined in prior figures, needs to be provided the first time this figure is referenced in the text rather than later.

5) Figure 3, as above a simpler indicator of light onsets and offsets would be better than the bulb/fiber illustrations.

6) Blocking of STN activity should induce involuntary movements. Did Halo-expressing mice show any abnormal behaviors?

7) Optogenetic activation of STN neurons interrupted bout of licking. Did it interrupt other behaviors?

For the points #6 and 7, we would like to see the authors' response if the authors already have a relevant data.

The original review comments from each reviewer are appended below:

*Reviewer #1:*

The authors have addressed most of the previous concerns. To confirm the effect of Arch-mediated inhibition of subthalamic nucleus neurons, the authors performed in vitro experiments. Although the data in vivo is still missing, this is an important addition.

*Reviewer #2:*

The manuscript has been greatly improved by additional data, analysis, and figures and is an important contribution to the literature on STN and behavioral inhibition.

*Reviewer #3:*

The authors examined self-initiated licking behaviors of mice during optogenetic activation or inhibition of the subthalamic nucleus (STN). They used mice whose STN neurons expressed specifically channelrhodopsin (ChR2) or halorhodopsin (Halo). Optogenetic activation of STN neurons interrupted bout of licking. Inhibition of STN neurons decreased interruption of liking by surprise stimuli. They consider that the STN is necessary and sufficient for suppression of behaviors.

1) The authors did not show any clear evidence that yellow laser inhibited STN neurons or their targets in vivo.

2) Blocking of STN activity should induce involuntary movements. Did Halo-expressing mice show any abnormal behaviors?

3) Optogenetic activation of STN neurons interrupted bout of licking. Did it interrupt other behaviors?

---

## [Author Response]

[Editors’ note: the author responses to the first round of peer review follow.]

*Essential points:*

*1) Although the authors confirmed the effect of optogenetic activations via channelrhodopsin-2 (ChR2), they do not show any demonstration that halorhodopsin inhibited the activity of neurons in the subthalamic nucleus (STN). This is an important point to demonstrate in order to interpret the data. Were STN neurons purely inhibited? What about rebound excitations? Please address this issue by in vivo recording.*

Note: Our lab lacks the capability to perform in vivo recordings. Indeed, single-unit recordings from phototagged neurons in mice remains a comparatively specialized feat. We thus proposed an ex vivo assessment – and *eLife* Editors provisionally ratified this strategy via correspondence in December 2016.

We performed systematic ex vivo (acute brain slice) recordings from STN neurons expressing Halorhodopsin:YFP. To mimic our in vivo conditions we inhibited STN neurons for 1 s – and we now see rapid and nearly complete silencing of spontaneous activity during this time. However, we also observe rebound excitation, as predicted by the reviewer, that appears to result in 200-300 ms of increased firing. These data are displayed in Figure 3.

The core question, however, is how might this rebound excitation impact our behavioral data (which show that STN inhibition blunts the effect interruptive effects of surprise)? We think very little or not at all. First, in our behavioral task, the second lick in a bout triggers the laser, and then the surprise stimuli occur after a 50-ms delay. Our dependent measure is whether more than three licks were made within the same bout.

On any given bout this fourth lick would occur hundreds of ms before the laser turns off. Thus any rebound excitation that might occur in vivo would occur only well after our dependent measure. Second, based on our results showing that ChR2 activation of the STN is sufficient to interrupt behavior (Figure 2), we would expect rebound excitation to have the opposite effect, potentiating, rather than reducing, the interruption we observe in response to surprise. Third, absent surprise, 1-s Halo inhibition of the STN had no effect on licking (Figure 3), suggesting any rebound excitation that might occur in vivo was insufficient to alter this behavior.

*2) In Figure 1, the authors show the effect ChR2 stimulation in the STN. Spiking of the STN neurons seems essentially continuous during the stimulation train. Does this indicate that synchronous activation of STN neurons caused persistent reverberatory activity? Please show the data after the last pulse as well so that we can see the extent at which the effect of stimulation lasts after the termination of the stimulation.*

We repeated these experiments to better align with the stimulus we used in our behavior, i.e., using a single 50-ms pulse, or a train of 10 pulses at 40 Hz. We observed no evidence of persistent reverberatory activity. Rather there is some apparent postexcitatory rebound inhibition. These revised data have been analyzed and plotted in Figure 1.

*3) Figure 1. If the idea is that STN can rapidly arrest behavior, why show target firing rate averaged over a 5s stimulation period? Presumably any relevant effect on target firing must be occurring within less than ~50ms or so – please show the time courses of target firing shortly after light onset instead.*

Indeed, our data suggest the postsynaptic responses are quite fast. We provide example traces and have made histograms to illustrate this point (Figure 1—figure supplement 1). Because the neurons were typically firing at less than 20-Hz we used bin sizes of 200 ms in these histograms, shorter bins (e.g., 50 ms) often lack any AP and are thus more variable. The EPSCs also indicate that the excitatory effects begin immediately upon blue light pulses (Figure 1—figure supplement 1); latency of EPSCs in response to light: STN^→SNr^: 1.03 ± 0.03 ms; STN^→GPe^: 1.03 ± 0.05 ms; STN^→EP^: 1.00 ± 0.20 ms.

*4) In general, it would be important to show less processed behavioral data, which may give a clearer view of what is going on. For instance, in Figure 2, the authors show an idealized schematic of a regular lick bout, and then information on lick bout length. Please show actual lick bout timing. Examples of real bouts may be helpful, and also simply rasters and histograms of lick density over time (as if licks were spikes).*

We have revised figure schematics to include example lick data (Figure 2, Figure 3). We have added examples rasters and histograms for both ChR2 interruption in Figure 2 – and Halo inhibition of surprise interruption in Figure 3.

*Also, are the bouts very distinct from each other or is the mouse essentially licking all the time with occasional >750ms gaps that define bout boundaries?*

Lick bouts were defined using an interlick interval threshold of <750ms based on our rough assessment of naturalistic licking patterns. These data are now included in Figure 2—figure supplement 2. In Figure 2—figure supplement 2 we also include a raster plot illustrating raw lick timestamps over an entire 30-min session.

*5) Figure 3. For the surprise experiment, why use such a long laser stimulation pulse (1s) given that any relevant neural effects must be much faster? Is this related to the definition of a "bout" as involving <750ms inter-lick-intervals?*

Because the interruptive effects of surprise do not persist indefinitely (Figure 3—figure supplement 2), we initially chose not to use the same cohort of mice to test multiple conditions. Thus, we had conducted the surprise experiment only once per cohort and using the described conditions (1-s inhibition initiated 50 ms prior to the surprise). Our rationale for choosing the 1-s period of inhibition follows: 1) We thought it prudent to start the inhibition just prior (50 ms) to surprise onset to ensure that STN was maximally inhibited by Halo. 2) We thought it important to sustain the inhibition for the duration of the surprise stimulus (500 ms). 3) We estimated that it would take 100-300 ms for surprise to activate the STN based on physiological measurements in humans (Wessel & Aron, 2013, Wessel et al., 2016). 4) We could not be certain how long the putative surprise-driven increase of STN activity would persist, but rodent STN responses to stop signals persist for ~50 ms (Schmidt et al., 2013). Thus 1-second inhibition was selected by summing the times 50+500+300+50 =900 ms; and rounding this up to 1 second.

In hindsight, it is also fortunate timing given the potential for post-inhibitory rebound to occur. By delaying the potential for post-inhibitory rebound until 950 ms after surprise onset, we can be confident that the blunting effects on surprise occurred while the laser was still on, and were not a consequence of rebound.

Note: We have repeated this experiment in a new cohort of mice using the same conditions but blocking light leakage. We also tested the effects of inhibiting STN for longer prior to surprise onset, as detailed in response to point #7 below.

*In addition, it seems that it took a long time for STN neurons to be inhibited by the halorhodopsin-based inhibition, compared with the excitation by ChR2-based stimulation. This also points to the importance of characterizing the effect of halorhodopsin-based inhibition on STN neurons (point #1).*

We did not provide data on how rapidly STN neurons were inhibited by Halo in the original submission. We now include these data and see photocurrent onset is essentially immediate (<1ms) and inhibition of spontaneous firing is evident within the first 100 ms (Figure 3).

*6) Figure 2—figure supplement 1. Why would green light alone (YFP group) tend to interrupt behavior when such an effect was not seen for blue light alone (YFP group) in the previous experiment?*

We cannot be certain, but it may be because the green light is on for a total of 1s while the blue light is only on for 5% of 1 s (50 ms pulse) or 10% of 1 s (10 pulses at 10 ms pulse width). Visually (to human eye) the light leak from the green pulse was brighter and more reflective in the operant box.

We have since developed a strategy to eliminate all light leakage and have replicated the Halo effect on surprise-induced interruption with a new cohort of mice. See response to point #7below.

Note: The movies submitted with manuscript were made without light guards – so that light pulses could serve as a visual aid to viewer, but data acquisition was always conducted with light guards to block the majority of the light leakage from fiber couplers.

*7) Halo and YFP animals showed different lengths of lick bouts in the baseline (no surprise, no laser condition, although the difference was not significant) (Figure 3). Although surprise resulted in similar lick bout lengths in Halo and YFP mice (Figure 3), the difference in the baseline might be problematic. Combining these results, the main result in Figure 3 could be explained by a mixture of laser itself, individual biases, and the effect that the authors are looking for (the role of STN). The authors must discuss this.*

To address this concern and eliminate the confound of behavioral interruption caused by the surprising/distracting effects of light leakage itself, we now repeated the experiment blocking all light leakage in a new cohort of mice. The main effect of STN inhibition on surprise-induced interruption persists; and in this experiment we observed no hint of difference between the groups under baseline conditions. Thus, absent light leakage we see no effect of light in the YFP group. We have thus replaced all data in Figure 3 with data from this new experimental cohort (cohort #3 below). Moreover, a compilation of the datasets across all conditions is appended as Figure 4.

Author response image 1.^†^These data included in the revised manuscript.(3^rd^ cohort was implanted to eliminate the interruptive effect of laser light leakage from the head cap observed in previous experiments) ^‡^These data included in previous submission.**DOI:**
http://dx.doi.org/10.7554/eLife.27689.013

*Reviewer #1:*

*1) The authors only examined the effect of STN optogenetic activation by* in vitro *recording and c-fos immunohistochemistry. Results of* in vivo *electrophysiological recordings, how STN neurons and their targets are activated by light through the fiber optics placed above the STN, are necessary.*

Single-unit in vivo recordings from phototagged mouse STN neurons are beyond our present capability. See response to Essential point #1.

*2) Moreover, they did not show any in vivo and in vitro electrophysiological results during STN silencing by Halorhodopsin. These data are indispensable.*

Ex vivo recordings from STN neurons expressing Halo are now included in Figure 3 – See response to Essential point #1.

*3) How effective is the activation or inhibition of STN neurons? Silencing of the STN induces motor abnormal behaviors such as hemiballism. Did animals show abnormal behaviors, such as hemiballism or rotational behaviors during strong silencing of the STN?*

All neurons recorded from ex vivo showed a response. No motor effects were apparent with 1-sec unilateral or bilateral Halo inhibition. Though beyond the scope of our study, these data may indicate that STN-related hemiballism may reflect compensatory changes resulting from sustained inactivity.

*4) Why did authors use different vectors between behavioral experiments and* in vitro *electrophysiological experiments? They should use the same vectors and examine the effectiveness by electrophysiological methods.*

We generally prefer to use mCherry for ex vivo work, so that we can avoid shining blue light on cells (massively driving their activity) while we look for fluorescent neurons to patch. In addition, we find it easier to identify soma expressing mCherry:opsin compared to YFP:opsins (perhaps because the YFP variants disperse more readily to distal processes). We generally prefer to use YFP for our in vivo work because we have observed mCherry photoconversion, and its emission/excitation spectra place greater limits on our ability to conduct multi-label immunostaining.

Nonetheless, for our newly performed electrophysiology experiments described in responses to Essential points #1 and #2, we used the same YFP vectors as those used in vivo. Our Methods subsection “Stereotactic surgery” has been modified appropriately.

*Reviewer #2:*

*To study the function of STN, the authors used a self-initiated bout of licking as an ongoing behavior that may be modulated by STN. This is a great choice because the behavior is natural and repeatable without any learning. The role of STN was examined by local activation and inactivation of STN neurons using optogenetic stimulation. Both of the data are critical for the conclusion that "STN is both necessary and sufficient for such forms of behavioral response suppression." The effects of STN activation are clear and convincing, but I have a question about the effects of STN inactivation, as shown below.*

*There is no clear evidence that photostimulation of STN in Halo-expressing mice inactivated STN neurons or their target neurons (GPe/EP/SNr), unlike the data shown for ChR2-expressing mice shown in Figure 1. This may be a bit tricky because STN neurons must be spontaneously active to see any effect on the target neurons. But if a prolonged stimulation (e.g., 1 s) is used (as in the behavioral experiment shown in Figure 3), the firing rates of the target neurons should decrease. Such data are important to proceed to the behavioral experiment.*

We have performed experiments showing rapid photoinhibition of STN neurons expressing Halo in response to light – Figure 3 and Response to Essential point #1.

Because STN soma are severed from their distal terminals, it seems unlikely that any spontaneous release from the Halo-expressing STN terminals would contribute to firing in postsynaptic GPe/EP/SNr firing in the ex vivo slice preparation.

I have two specific questions.

*First, how quickly can STN neurons suppress ongoing behavior? This is important to adapt to the rapidly changing environment. To address this question, I have been checking the data in Figure 2. I assume that the blue window indicates the stimulation period. The effect of ChR2-laser diverged from the control around 100 ms after the onset of the stimulation. This looks explicit, but I am not convinced. My understanding is that the data lines are shown relative to the total number of licks within a bout of licks. Since the number of licks was smaller in ChR2-laser trials, the data lines are not presented based on the actual frequency of licks. I would simply show the cumulative number of clicks. Then, the blue line (ChR2-laser) would be lower, and the differentiation latency may be shorter than 100 ms. Another reason for asking this question is that the effects of the photostimulation on the STN-target neurons in GPe, EP, and SNr are fairly quick (Figure 1) (although I cannot see the actual latencies). If these target neurons respond, say, in 2-3 ms, I expect that the behavior would be suppressed much earlier than 100 ms.*

In fact the graphs plot the cumulative frequency by trial type – and only for bouts that included a third lick. Thus, these data reflect only bouts that were not interrupted. On the minority of bouts that activation of STN doesn’t interrupt the bout (as defined by our <750 ILI criteria for defining a bout), STN activation still imposes a ‘pause’.

The reviewer is correct that this pause does not appear to manifest prior to ~100ms, which fits with the natural rate of licking and suggests that if mice happen to be licking faster than ~10Hz, an STN command to Stop/pause/interrupt is too slow to prevent the execution of the subsequent lick.

The line plots in Figure 2 (formerly Figure 2) are shown relative to the total number of bouts of that trial type, thus are normalized for the differences in the total number of 2^nd^-3^rd^ interlick interval values across trial types. We’ve now changed the Y-axis on revised Figure 2 to “cumulative proportion (by trial type)” to help clarify.

*Second, I have some questions about the data showing the effect of the inactivation of STN neurons (Figure 3). According to my understanding, the photostimulation started simultaneously with the 2nd lick, and then the surprising event started after 50 ms. Is this because the authors had tried several versions and found that this temporal order was most effective? Apparently, it took a long time for STN neurons to be inhibited by this Halo-based stimulation, compared with the excitation by ChR2-based stimulation. Relevant to this question: What was the latency of the behavioral suppression in response to the surprising event?*

See responses to Essential point #1 for the timecourse of Halo inhibition ex vivo; and Essential point #5 for an explanation of the logic behind the temporal order for the Halo surprise experiment. We can’t measure the latency to behavioral suppression, which would represent *the absence* of a lick event, but Figure 3 provides an example comparing the timing of licks on no stimulus and surprise trials.

*Other specific questions and comments:*

*Subsection “Optogenetic activation of STN excites output nuclei”, last paragraph: How did you define 'postsynaptic cells'?*

Post-synaptic cells were neurons located within the GPe/SNr/EP that were juxtaposed to fluorescence from STN terminals and that showed a short-latency EPSC upon photostimulation (subsection “Optogenetic activation of STN excites output nuclei”).

*In Figure 1, please indicate the time and EPSC amplitude for the example data. What was the latency of EPSC in response to the stimulation?*

Scale bars are included in what is now Figure 1—figure supplement 1. The latency of EPSC in response to light for each target is: STN^→SNr^: 1.03 ± 0.03 ms; STN^→GPe^: 1.03 ± 0.05 ms; STN^→EP^: 1.00 ± 0.20 ms.

*In Figure 1—figure supplement 1, does 'ChR2 excluded' mean that the data obtained with these stimulation sites were excluded? I presume that the stimulation effect was absent or weaker than the others. Such data may be important to support the conclusion: the stimulation affected STN, not other areas.*

We understand, but have not the sample power to make such claims. The two mice were excluded by observers blind to behavioral results, solely because the fiber tip appeared to be placed in excess of 0.5 mm of the STN. However, expression of ChR2:YFP in the STN was acceptable, and it seems probable that some light made it to the STN. In Figure 5 we provide behavioral data across all conditions as well as histology images for both excluded mice in the ChR2 study for your inspection (note: scalebars are 0.5 mm). Importantly, no conclusions would change were we to include the excluded mice in the analysis.

Author response image 2.**DOI:**
http://dx.doi.org/10.7554/eLife.27689.014

*Data in Figure 1—figure supplement 2. Was the stimulation intensity 0.5mW or 10mW?*

All the images are from the 10-mW condition, we edited the legend to reflect this.

*There are some c-Fos labeled cells outside the presumed target areas. For example, I wonder if labeled cells outside SNr (K) are located in VTA.*

We have revisited our slides and it is not clear that Fos is induced in the VTA, but there may be some Fos induction in the substantia nigra compacta.

*Please indicate how many animals were used for each experiment.*

For Fos cell counts in panel E (now Figure 1), we used n=3 ChR2 mice for the 10mW condition, n=4 ChR2 for the 0.5mW condition and n=4 YFP mice. We have added these values to the figure legend.

*Figure 2 indicate that the total number of bouts increased in ChR2-laser condition. Does this mean that the total number of licks increased? Any interpretation?*

Our previous analysis included across-group comparisons of total number of bouts (laser and nonlaser trials included), which showed an increase in the total bouts in the ChR2 compared with YFP (now Figure 2—figure supplement 1). This may reflect compensatory drinking to “make up for” the interrupted reward consumption during laser trials by licking more during non-stim trials. We also see a significant increase in the total number of licks for the 50-ms pulse condition (Figure 2—figure supplement 1).

*Figure 2—figure supplement 1 shows that the bilateral stimulation was less effective. Any reason?*

We do not believe that there is a significant difference between bilateral and unilateral groups, though our study was not designed to test that possibility, and the small group size for the bilateral cohort (n=5) would make it difficult to make such a claim. We only conclude that both unilateral and bilateral stimulation is sufficient to interrupt licking.

It is interesting to note that licking is not a particularly ‘lateralized’ behavior.

*Figure 3—figure supplement 1 suggests the non-selective effect of photostimulation which seems to act as another surprising event. Is it difficult to block the light from the head cap?*

See responses to Essential points #6 and #7.

*Figure 3—figure supplement 2 indicates that the behavioral suppression became weaker as the surprising event was repeated. I wonder if this is caused by the decrease in the sensitivity of STN neurons to the surprising event.*

This is an interesting hypothesis that our present experiments do not illuminate.

*Was the experiment shown in Figure 3 started after the habituation shown in Figure 3—figure supplement 2? If so, why?*

The data shown in Figure 3 were and are from mice naïve to the ‘surprise’ stimulus. Figure 3—figure supplement 2 was done in a separate cohort of untreated control mice to a) validate our paradigm and b) to show that the effects of surprise wear off gradually across session, rather than, say, rapidly within session.

*"Importantly, in the absence of the sound/light event we found that STN inhibition did not alter licking behavior compared to the YFP controls."*

*This is important, but I cannot find data.*

See Figure 3—figure supplement 1, which has now been revised and updated in the resubmission.

*While reading the manuscript, I had a difficulty in finding which video I should check.*

We have properly cited the movies in the text.

*Reviewer #3:*

*In this brief report Fife et al. present optogenetic results supporting the idea that the STN is involved in interrupting ongoing behavior; activation of STN tends to interrupt bouts of licking, while suppression of STN tends to prevent interruption of licking by surprising cues. Though limited in scope these results based on manipulations are a useful complement to the extensive literature on STN & stopping based on correlations. But it would be good to show less processed behavioral data, which may give a clearer view of what is going on.*

*1) (Figure 1) Spiking of the STN neurons seems essentially continuous during the stimulation train. This seems a bit strange – is synchronous stimulation of STN neurons causing persistent reverberatory activity? Please show us when it stops after the last pulse. In any case "spikes/stimulus" seems like an inappropriate measure, since it's not clear which spikes are being evoked by which stimulus.*

See response to Essential point #2. Also, we have modified the graph to show% change in firing rate relative to baseline.

*2) (Figure 1). If the idea is that STN can rapidly arrest behavior, why show target firing rate averaged over a 5s stimulation period? Presumably any relevant effect on target firing must be occurring within less than ~50ms or so – please show the time courses of target firing shortly after light onset instead.*

*S*ee response to Essential point #3.

*3) (More on Figure 1). Figure 1: probably unnecessary these days. Figure 1: not clear what is being shown – is this supposed to be ChR2 expression in STN cell bodies and in fibers (only) within STN targets? Probably better just to use Figure 1—figure supplement 1 as a main figure instead. Figure 1: scale bar scales are not shown or given in caption.*

We have modified Figure 1 and re-organized the data included in supplemental. The data in Figure 1 now deals primarily with STN cell bodies, while Figure 1—figure supplement 1 deals with postsynaptic cells in GPe, EP, and SNr.

*4) (Figure 2) We are shown an idealized schematic of a regular lick bout, and then information on lick bout length, but please show actual lick bout timing. Examples of real bouts may be helpful, and also simply rasters and histograms of lick density over time (as if licks were spikes). Also, are the bouts very distinct from each other or is the mouse essentially licking all the time with occasional >750ms gaps that define bout boundaries?*

We have made considerable changes along these lines, see response to Essential point #4.

*5) (Figure 2) If STN arrests licking, but mice adjust by increasing the number of bouts, how long does the arrest last?*

We have now assessed the interbout interval following laser and nonlaser trials (Figure 2—figure supplement 1). The arrest in licking following STN-mediated interruption is longer lasting as described in the Results (subsection “Activation of STN interrupts or pauses behavior”, last paragraph).

*6) (Figure 2) How fast is the closed-loop control? I.e. what is the time from 2nd lick onset to laser pulse onset?*

From the behavioral data we can infer that it is less than the normal 100-150 ms interlick interval. But to get at this question more directly, we captured high-speed, high resolution movies and measured the time between laser ON relative to lickometer contact. Analysis of 3 identically filmed videos revealed no detectable delay between film frames showing visible contact with the lickometer and laser light emerging from the optic fiber cable. With a resolution of 240 fps, the control loop is thus <4.2 ms.

*7) (Figure 3) For the surprise experiment, why use such a long laser stimulation pulse (1s) given that any relevant neural effects must be much faster? Is this related to the definition of a "bout" as involving <750ms inter-lick-intervals?*

See response to Essential point #5.

*8) (Figure 3) Schematic – it's not very clear what is what.*

The schematic has been revised for clarity (Figure 3).

*9) (Figure 2—figure supplement 1) Why would green light alone (YFP group) tend to interrupt behavior when such an effect was not seen for blue light alone (YFP group) in the previous experiment?*

See response to Essential points #6 and #7.

*10) Discussion is a bit cursory. For example it would have been helpful to discuss the fact that the results here are based on interrupting an already-started action (if this is a fair description) compared to canonical results based on preventing action initiation at all.*

We have added this important discussion point (Discussion, fourth paragraph), and several others.

[Editors' note: the author responses to the re-review follow.]

*Essential points:*

*1) Figure 1: It's not entirely clear what point is being made with the zoomed-in insets, and what the nuclear marker is in the YFPctrl inset.*

These data were requested by *eLife* in pre-review. They show expression at the cellular rather than regional level. The nuclear marker NeuN is used in all insets, including the YFP control. We now indicate this in updated panels 1A-D and legend.

*2) Figure 1—figure supplement 1 not referenced in text?*

Figure 1—figure supplement 1 is referenced in the subsection “Optogenetic activation of STN excites output nuclei”.

*3) Figure 2. The illustration of blue light coming out of fibers to interrupt licking is pretty, but better would be an unequivocal mark of exactly the period of blue light illumination.*

A simplified scheme is included in revised Figure 2 and now includes a clear and precise signifier of blue light illumination relative to lick timestamps.

*4) Figure 3. The explanation for using "% 3-lick-bouts", rather than 2-lick-bouts as examined in prior figures, needs to be provided the first time this figure is referenced in the text rather than later.*

We have adjusted Figure 3 to report% 2-lick bouts, since this experiment has no delay followed by surprise. This does not change the conclusion, which is that 1-sec Halo- inhibition of STN on its own did not alter licking, and to support this claim we have added the% 3-lick,% 5-lick, and% 10-lick-bout data to Figure 3—figure supplement 1, showing the persistence of the negative effect irrespective of the dependent measure.

The results now explain reason for% 3-lick bouts on first use as it relates to Figure 3.

*5) Figure 3, as above a simpler indicator of light onsets and offsets would be better than the bulb/fiber illustrations.*

A simplified scheme is included in revised Figure 3, to parallel the newly designed Figure 2, and includes a precise signifier of green light illumination relative to lick timestamps.

*6) Blocking of STN activity should induce involuntary movements. Did Halo-expressing mice show any abnormal behaviors?*

1-s Halo-inhibition of STN did *not* induce noticeable involuntary movements. Though we did not test this explicitly, we note that licking was unaltered by 1-s Halo-inhibition of the STN, e.g., Figure 3 and Figure 3—figure supplement 1. We now comment on this in the Discussion (second paragraph), suggesting that hemiballismus associated with STN lesions may reflect compensatory changes in basal ganglia circuits following more sustained perturbations of STN activity.

*7) Optogenetic activation of STN neurons interrupted bout of licking. Did it interrupt other behaviors?*

We used licking because it is a directed behavior that we can measure with high precision. Though we aim to develop other suitable behavioral assays, we have no relevant data to add at this time.

Addendum: While reviewing all of our datasets in preparation for final file submission we discovered data transposition errors in three animals included in prior Figure 3. After correcting the errors, a statistical interaction (comparing the effect of Halo/YFP treatment groups by laser/surprise stimulus condition) became just non-significant. We are happy to provide additional details on this honest mistake upon request.

However, this is an experiment we repeated 5 times on 3 cohorts of animals. Though the effect did not always reach significance, in each of these five experiments it showed the same trend. Thus, we have swapped Figure 3 with the dataset in previous Figure 3—figure supplement 1, included the corrected dataset in Figure 3—figure supplement 1, where we also now include data from all 5 experiments. Finally, while keeping an example raster in Figure 3, we replaced the binned histogram plots in Figure 3 that had showed only an example mouse, with cumulative probability plots that display the entire dataset for the experiment. After much consideration we feel this is the most transparent and rigorous response.